# Morphological Characteristics, Ultrastructure, and Chemical Constituents of the Endotesta in Ginkgo (*Ginkgo biloba* L.)

**DOI:** 10.3390/plants12203560

**Published:** 2023-10-13

**Authors:** Fangdi Li, Ganping Liu, Linying Zhao, Xiaoge Gao, Zhuolong Shen, Fuliang Cao, Qirong Guo

**Affiliations:** 1Co-Innovation Center for Sustainable Forestry in Southern China, College of Forestry, Nanjing Forestry University, Nanjing 210037, China; lfd@njfu.edu.cn (F.L.); xiaoliu@njfu.edu.cn (G.L.); gaoxiaoge@njfu.edu.cn (X.G.); szl@njfu.edu.cn (Z.S.); caofl@njfu.edu.cn (F.C.); 2Suzhou Planning and Design Research Institute Co., Ltd., Suzhou 215000, China

**Keywords:** chemical composition, development, Ginkgo endotesta, membranous structure

## Abstract

*Ginkgo biloba* L. is a tree species of significant economic and ecological importance. Prior studies of the *Ginkgo biloba* seed coat have predominantly focused on the sarcotesta and sclerotesta, with less attention paid to the endotesta. In this study, the development and formation of Ginkgo endotesta were examined using light microscopy and transmission electron microscopy. The structural properties of the mature endotesta were analyzed using micro-CT imaging and scanning electron microscopy. The results indicate that the endotesta possess a membranous structure primarily originating from the inner bead peridium, a segment of bead core tissue, and the macrospore membrane. The endotesta from the middle constriction line to the chalazal end comprises a single layer with a greyish-white papery structure. In contrast, the endotesta was divided into two inner and two outer layers, from the middle constriction line to the micropylar end. The outer endosperm adheres closely to the sclerotesta, while the inner endosperm adheres to the seed kernel. The surface of the endotesta was irregularly raised, with thicker wax at the chalazal end, whereas the micropylar end demonstrated similar characteristics with thinner wax and tumor layers. The endotesta contained 17 amino acids, 18 fatty acids, 10 trace elements, and 7 vitamins. Overall, its nutritional value was relatively well balanced.

## 1. Introduction

*Ginkgo biloba* L., an ancient tree species originating in China, is the sole surviving Ginkgoales within gymnosperms [1]. The development of the naked ovule of *Ginkgo biloba* directly leads to seed formation. The mature *Ginkgo biloba* seed comprises five main parts: the sarcotesta, sclerotesta, endotesta, endosperm, and the seed embryo. Among these, the endosperm and seed embryo are often collectively referred to as the seed kernel [2,3,4].

The seed coat of *Ginkgo biloba* develops from a single peridium, which is a single layer near the center of the bead [5]. It consists of three distinct layers, each exhibiting significant differences in morphology and structure. The sarcotesta which appears fleshy and orange-yellow, has inconspicuous dorsal and ventral surface. The sclerotesta is bony, hard, and whitish, displaying a thicker and slightly rougher woody texture at the chalazal end that is resistant to cracking, whereas the micropylar end is thinner, with a uniformly smooth texture susceptible to cracking. The endotesta has a membranous nature, and notable variations exist in morphology and structure between its upper and lower parts, demarcated by the middle constriction line. The upper part extends from the middle constriction line to the chalazal end, and appears opaque greyish-white. In contrast. the lower part from the middle constriction line to the micropylar end is divided into two layers, one adhering to the sclerotesta and the other to the seed kernel, both exhibiting semi-transparency and displaying a golden-brown coloration [6,7,8,9,10,11].

Traditionally, the bead peridium tissue is categorized into outer, middle, and inner bead peridium layers, corresponding to the formation of the sarcotesta, sclerotesta, and endotesta of *Ginkgo biloba* [12,13,14]. However, the development of the endotesta remains poorly studied. In recent years, existing studies have indicated that the micropylar end of the inner zone of the bead peridium exhibits a cone-like structure that is distinct from the bead peridium [15,16]. This cone-shaped structure is located on the inner side of the peridium at the micropylar end of the Ginkgo ovule. Its profile is symmetrical, light-green, and translucent. Carothers first described it in an article on the development of the female gametophyte of the Ginkgo ovule, though comprehensive investigation is lacking [17].

The developmental process of the endotesta is closely related to the peridium anatomy. Favre-Duchartre documented the Ginkgo developmental cycle and briefly discussed the periderm development [18]. Takaso investigated the development of Ginkgo peridium and compared it with other gymnosperms [19]. Since then, scholars have further studied, summarized, and reviewed the three layers of *Ginkgo biloba* seed coat. Taking the differentiation of periderm tissue cells (mid-May) as the boundary, the periderm is referred to as “periderm” before this point and the “seed coat” thereafter; or taking Ginkgo ovule fertilization (end of August, beginning of September) as the boundary, the pre-fertilization structure is called “periderm”, and post-fertilization it is called the “seed coat”. “Periderm” encompasses the entire developmental and formational process [18]. The membranous endotesta of *Ginkgo biloba* hinders water dissipation, effectively preventing water loss and desiccation of the seed kernel, thereby ensuring the quality of the white fruit [16]. Meanwhile, the presence of endotesta isolates the seed kernel during early development from the external environment, guaranteeing the integrity and independence of the developmental processes of the seed embryo and endosperm.

Ginkgo seed kernels, devoid of the exocarp and mesocarp, possess substantial medicinal and nutritional value. Conversely, exocarps and mesocarps, often treated as by-products, are frequently discarded as waste. Ginkgo exocarps contain flavonoids, terpene lactones, fats, polysaccharides, phenolic acids and other compounds suitable for eco-friendly biopesticide development, medicines targeting cardiovascular ailments, and health promoting formulations such as Ginkgo exocarp polysaccharide capsules. Ginkgo mesocarps contain flavonoids, lactones, phenolic acids, glycerides and other constituents that can be augmented through optimized processing, thereby offering a novel avenue for harnessing Ginkgo mesocarp resources [20,21]. The endocarp situated as a thin film structure closely adhering to the seed kernel, is typically discarded during the production of Ginkgo canned food, juices, wines, and other products. Limited research currently exists on endotesta components, with only sugars, phenolic acids, and trace elements detected and analyzed [22].

In this study, we comprehensively investigated endotesta development through morphological observations, micro-CT imaging, scanning electron microscopy (SEM), and transmission electron microscopy (TEM), thereby facilitating a holistic comprehension of Ginkgo endotesta development and its underlying biological processes. Furthermore, employing high-performance liquid chromatography (HPLC), we analyzed the primary chemical components of *Ginkgo biloba* endotesta, including amino acids, fatty acids, vitamins, inorganic elements, and other major constituents. This analysis establishes a theoretical foundation and reference point for storing and preserving *Ginkgo biloba* seeds for fruit-based applications and for the resourceful utilization of the endotesta.

## 2. Results

### 2.1. Development of the Endotesta of Ginkgo biloba

#### 2.1.1. Morphological Observations of *Ginkgo biloba* Endotesta during Development

In mid-April (4.10–4.15), following pollination, the Ginkgo ovules exhibited rapid growth and thickening. Notably, the micropylar end revealed the presence of a discernible cone-like structure within the inner integument, with evident separation from the nucellar tissue. In contrast, at the chalazal end, the inner integument remained unseparated from the nucellar tissue (Figure 1a). In late April (4.25–4.31), enlargement and thickening of the outer integument was observed, accompanied by differentiation of the middle integument (Figure 1b). In early May (5.1–5.5), the structure of the chalazal-end integuments became distinct, with each undergoing noticeable differentiation. The outer integument displayed thickness alongside the secretory cavities, and the middle integument exhibited a thin yellowish character. The inner integument situated between the outer and middle integuments, demonstrated a creamy white hue with a light green hint. As ovule development continued, the ‘conoid’ structure embedded within the middle integument, comprising about one-third of its volume, progressed to form the outer endotesta at the micropylar end. Concurrently, the nucellar tissue underwent degeneration, mostly vanishing while leaving remnants at the micropylar end. Together with the inner integument, the nucellar tissue at the chalazal end contributed to the formation of the chalazal endotesta (Figure 1c). In mid-May (5.10–5.15), longitudinal growth of the middle integument coincided with the thickening of the inner integument and nucellar tissues at the chalazal end. The ‘conoid’ structures expanded laterally to their maximum and initiated longitudinal elongation (Figure 1d–f). Advancing to early June (6.1–6.10), the middle integument’s growth ceased as lignification commenced, while the ‘conoid’ structure experienced compression and began to degenerate into a narrow band (Figure 1g,h). In late June (6.25–6.30), lignification of the middle integument became evident, causing the ‘conoid’ structure to narrow significantly, alongside a noticeable reduction in the width of the inner integument at the chalazal end (Figure 1i–k). In early July (7.1–7.5), the middle integument exhibited clear lignification, rendering it resistant to cutting. At the chalazal end, the inner integument displayed thinning, while the previously transparent ‘conoid’ structure entirely degraded and vanished (Figure 1).

#### 2.1.2. Microstructure of the Endotesta during Development

By mid-April, post-pollination, discernible differentiation had occurred within the Ginkgo ovules, resulting in distinct integument and nucellar tissues. The cells of the outer integument were larger and had a higher prevalence of secretory cavities. Conversely, the cells of the middle integument were comparatively smaller while featuring larger nuclei. At the chalazal end of the inner integument, cell connectivity with the nucellar tissue was established. Notably, both cells were of similar dimensions. At the micropylar end, a ‘conoid’ structure emerged comprising thin-walled cells arranged in an irregular polygonal manner. These cells were closely arranged and displayed well-defined boundaries separating them from the adjacent nucellar tissue (Figure 2a,b).

In early May, conspicuous gaps were apparent within the cells of the inner integument at the chalazal end, indicating heightened cell division. These cells increased in size, had an irregular configuration, and contained granular inclusions. Large nuclei positioned proximal to the cell walls were also evident. Concurrently, the ‘conoid’ structure cells within the inner integument at the micropylar end continued to divide, displaying an irregular polygonal shape and dense arrangement. Several cells contained granular inclusions. The innermost cells of the inner integument transitioned towards a rounded or squared morphology, effectively distinguishing them from the nucellar tissues. Conversely, the outermost monolayer of cells underwent elongation (Figure 2c). In late May (5.25–5.31), the inner integument cells at the chalazal end exhibited a densely packed arrangement, with an increase in size. These cells assumed a long elliptical shape, with the granular material within them presenting a deeper coloration. In contrast, the ‘conoid’ structure cells within the inner integument at the micropylar end underwent thickening and rapid expansion in size. They displayed an irregular shape and were sparsely arranged, with only a limited number of nuclei. As the nucellar tissue underwent degeneration and thinning, the cell size diminished and the outer cells were extended longitudinally. In particular, the inclusions of the cells at the demarcation line between the inner integument and the nucellar tissue exhibited deeper staining (Figure 2d).

In early June, the gaps between the cells of the inner integument at the chalazal end became distinctly more discernible, assuming a narrow or round configuration. Notably, the nuclei drew closer to the inner cell wall, displaying a proclivity for lysis. Simultaneously, ‘conoid’ structure cells within the inner integument at the micropylar end experienced compression due to the expanding female gametophyte, rendering them flattened and irregularly narrow with a loose arrangement. Meanwhile, the emergence of the megaspore membrane was accompanied by the gradual disintegration of nucellar tissue cells adjacent to the female gametophyte. This led to a loosely arranged, extremely irregular cellular morphology. Nucellar tissue cells proximal to the inner integument acquired a long oval or narrow shape, adopting a closely packed arrangement. Certain cells displayed a large nucleus accompanied by the accumulation of starch grains along the inner wall (Figure 2e). In mid-June (6.10–6.15), gradual lignification of the middle integument occurred. The inner integument cells at the chalazal end demonstrated a narrow, tightly arranged disposition. Conversely, the remaining cells displayed a loose arrangement coupled with variable shapes, whereas the nuclei began to disintegrate. The ‘conoid’ structure cells at the micropylar end expanded in size and adopted a narrow, elongated morphology (Figure 2f). In late June, cells at the chalazal end remained in an irregular arrangement, marked by significant differences in cell size (Figure 2g). Cells within the ‘conoid’ structures at the micropylar end became narrower and more irregular in shape. Cells adjacent to the middle integument exhibited partial apoptosis, with cell wall remnants discernible on the inner side of the middle integument (Figure 2h).

In early July, the cells within the inner integument at the chalazal end exhibited a loose arrangement, adopting round or oval shapes, and undergoing a reduction in size. Only a few cells contained lightly stained nuclei (Figure 3a). The ‘conoid’ structures within the inner integument at the micropylar end had developed into an outer inner integument consisting of 3–4 cell layers. Cells adjacent to the middle integument were grouped together, whereas cells in the middle layer displayed irregular shapes, variable sizes, and a closely packed disposition devoid of nuclei. The innermost layer of cells demonstrated elongated and closely arranged characteristics (Figure 3b). Cells near the female gametophyte exhibited degeneration, manifesting a narrow shape, accompanied by visible degenerative traces (Figure 3c). In mid-July (7.10–7.15), cells within the inner integument at the chalazal end exhibited an increased degree of looseness in their arrangement. Cells adjoining the middle integument underwent deformation, adopting a narrow configuration and initiating a gradual degenerative process. On the side facing the female gametophyte, nucellar cells also displayed a progressive trend towards cellular demise. Moreover, the cell wall remnants experienced thickening due to the superimposed layers (Figure 3d). The outer layer of the inner integument at the micropylar end underwent further degradation, resulting in the preservation of only two cell layers and a marked decrease in cell size (Figure 3e). The inner nucellar cells experienced accelerated degradation, adopting a highly irregular and narrow configuration. The outermost cells, in turn, degraded into flattened and elongated forms (Figure 3f).

In mid-August (8.1–8.5), the inner integument at the chalazal end featured 6–7 cell layers, displaying a loosely arranged disposition and reaching the final stages of disintegration (Figure 4a). Concurrently, the inner integument on the exterior side of the micropylar end underwent complete cellular demise, transforming into a membranous structure comprising multiple layers of densely clustered cell remnants (Figure 4b). Similarly, the inner nucellar tissue consisted of 4–5 layers of cells, characterized by a loosely arranged configuration and a crumpled irregular shape (Figure 4c). By late August (8.25–8.31), all cells within the nucellar portion of the micropylar end had died, resulting in the superimposition of cell wall remnants into bundles that were more loosely distributed (Figure 4d). By early September (9.1–9.5), these remnants had adopted a dense membranous structure similar to the exterior outer integument (Figure 4e).

By early September, the inner integument and nucellar cells situated at the chalazal end had undergone complete cellular demise. Multiple layers of residual filamentous cell walls were observed, sparsely arranged and enveloping dark cell inclusions between the cell walls (Figure 4f). In summary, Ginkgo endocarps developed from a combination of the inner integument and a portion of the nucellar tissue, rather than originating solely from the inner integument or the megaspore membrane.

#### 2.1.3. Ultrastructure of the Endocarp during Development

In late May, the cells within the inner integument at the chalazal end exhibited a regular alignment. These cells featured a notable nuclear size, dense cytoplasm, prominent nucleoli, and intact nuclear membranes (Figure 5a,b). Abundant cytoplasmic organelles were observed, primarily consisting of the endoplasmic reticulum, mitochondria and plastids (Figure 5c). The endoplasmic reticulum displayed a flattened and vesicular structure, with numerous granular ribosomes attached (Figure 5c). The mitochondria were abundant, displaying round, oval, and dumbbell shapes. These mitochondria contained abundant internal cristae and maintained an intact membrane structure. The plastids were ellipsoidal or orbicular in shape and characterized by a high electron density (Figure 5d). In addition, a few starch grains were present in the cells (Figure 5b).

The outer ‘conoid’ structures within the inner integument at the micropylar end exhibited an irregular arrangement. These structures appeared minor and highly vesicular, and were characterized by large nuclei, thick cytoplasm, distinct nucleoli and intact nuclear membranes (Figure 5e). The organelles within these structures were clustered towards the margins by a large central vesicle. Notable organelles included the endoplasmic reticulum, mitochondria, the golgiosome and plastids (Figure 5f). The endoplasmic reticulum maintained a flattened and vesicular configuration embellished with ribosomes. Abundant mitochondria were present, exhibiting round, oval, and elongated forms, with well-defined internal cristae. The golgiosome was constituted by numerous vesicles. The plastids had rod-shaped and ellipsoidal appearances and featured a high electron density (Figure 5g). Intercellular filaments were observed between adjacent cells (Figure 5h).

The inner ‘conoid’ structures located within the inner integument at the chalazal end were notably large. These structures possessed a uniform nucleoplasm, a well-defined nucleolus, an intact nuclear membrane, and small vesicles originating from the inner extensions of the cytoplasmic membrane (Figure 5i). Moreover, an increased number of rosary polyribosomes was observed (Figure 5j). The endoplasmic reticulum, mitochondria and golgiosome also featured prominently (Figure 5k). The endoplasmic reticulum exhibited an abundant distribution and was adorned with ribosomes (Figure 5l). The mitochondria displayed abundant inner cristae with an intact membrane structure (Figure 5m). The golgiosome was characterized by numerous flattened vesicles. Moreover, the cells were equipped with osmiophilic granules presenting a deeper electron density (Figure 5n), and higher concentration of starch granules, which manifested as spindle-shaped or elongated structures (Figure 5o).

In late June, notable nucleoplasmic crumpling was observed within the inner integument at the chalazal end (Figure 6a). There was a reduction in the inner cristae of the mitochondria, accompanied by a decrease in electron density, bilayer rupture, and the release of starch grains and other inclusions (Figure 6b). The endoplasmic reticulum was primarily near the cell wall (Figure 6b), with noticeable swelling of the rough endoplasmic reticulum (Figure 6c). Moreover, a greater quantity of significantly larger starch grains was identified, which displayed spherical or ellipsoidal shapes, along with pronounced folds of electron density on their surfaces (Figure 6d).

The outer ‘conoid’ structures of the inner integument in the cell at the micropylar end exhibited crumpling, accompanied by thickening of the cell wall (Figure 6e). Notably, the inner nuclear membrane ruptured, resulting in a sparse nucleoplasm and disappearance of the starch granules (Figure 6f). There was continued disintegration of the mitochondria leading to the disappearance of the inner cristae and the breakdown of the membrane structure (Figure 6g). During this phase of programmed cellular demise, degraded materials and organelle remnants were located near the inner cell wall, characterized by an extremely high electron density (Figure 6h).

The outermost layer of nucellar cells on the inner side of the integument at the micropylar end represented the macrospore membrane, with a thin and smooth texture. This membrane was characterized by numerous short and thick vertical lines. The subsequent 4–5 layers of cells demonstrated diminished cytoplasm content, along with remnants of thickened cell walls superimposed on each other. The inner layer of cells manifested obvious wrinkling, close arrangement, and irregular shapes, with a noticeable intracellular void (Figure 6i). The nucleoplasm of these nucellar cells condensed, leading to an internal protrusion of the plasma membrane and the formation of small vesicles (Figure 6j). Additionally, the endoplasmic reticulum presented a vesicular appearance while enclosing the organelle remnants. The mitochondria started to disintegrate, resulting in blurred inner cristae and rupture of the bilayer (Figure 6k). Starch granules were discovered in proximity to the cell wall, presenting a smooth surface (Figure 6l).

In late July (7.25–7.31), the cells within the inner integument at the chalazal end exhibited distinct crinkling and a close arrangement of elongated bands. The nuclei displayed a crinkled morphology, accompanied by visible nucleoli, ruptured nuclear membranes, and a sparse nucleoplasm (Figure 7a). The presence of numerous membrane vesicles near the cell membrane indicated a vascularized state in the endoplasmic reticulum (Figure 7b). Remarkably, the cytoplasm experienced a substantial reduction, whereas the cell wall experienced thickening, leading to an increase in electron density (Figure 7c). The inner cristae of the mitochondria were blurred, with evident bilayer ruptures (Figure 7d). The starch grains notably diminished and displayed variations in size (Figure 7e). The macrospore membrane on the outer side of the cell wall exhibited irregular flocculation (Figure 7f).

The outer layer of the inner integument cells predominantly experienced cellular demise, resulting in the formation of a membranous structure composed primarily of cell wall remnants.

The macrospore membrane of the outermost layer of the inner integument cell was aligned with the chalazal end. The cell wall of the middle layer became increasingly compact due to stacking; the number of cell layers in the inner layer decreased, resulting in crinkled and arranged cells with elevated cytoplasmic electron density (Figure 7g). The endoplasmic reticulum formed fragmented and vesicular structures in a ring shape, enveloping the mitochondria (Figure 7h). The mitochondria underwent gradual degradation, contributing to blurring the inner cristae bilayer ruptures, autolysis, and subsequent content leakage (Figure 7i). The surface starch granules exhibited predominantly parallel folding marks (Figure 7i).

### 2.2. Structure of the Endotesta

#### 2.2.1. Apparent Structure

The endotesta of mature Ginkgo seeds enveloped the seed kernel entirely, appearing thin and desiccated. This outer covering was divided along the longitudinal equatorial plane by the mid-constriction line, exhibiting distinct patterns above and below (Figure 8a). The endotesta adhered to the seed kernel from the mid-constriction line to the chalazal end, displaying a papery, texture and a dark brown or greyish-white coloration. Notably, two visible vascular bundle traces were discernible at the chalazal end (Figure 8b).

The endotesta spanning from the middle constriction line to the micropylar end exhibited a two-layered structure. The outer layer was in immediate proximity to the sclerotesta, whereas the inner layer adhered to the surface of the seed kernel. Both layers possessed a membranous and translucent russet appearance (Figure 8a,b), with a discernible creamy oval structure against the micropylar end (Figure 8c).

#### 2.2.2. X-ray 3D Micro CT Imaging

Based on the forementioned observations, the Ginkgo endotesta was segmented into three distinct regions: the inner integument at the chalazal end, outer endotesta at the micropylar end and inner endotesta at the micropylar end. These segments exhibited varying degrees of denseness within each section. To analyze the structural intricacies and denseness of each inner integument part, X-ray 3D Micro CT tomography was employed.

The findings revealed that the seed kernel in its natural state demonstrated a mild depression along the mid-constriction line (Figure 8d). The chalazal end comprised the morphological upper end, and the endotesta adopted a ‘herringbone’ shape (Figure 8e,f). Additionally, the endotesta at the chalazal end close to the seed kernel measured roughly 19.6 μm in thickness. This region exhibited a porous consistency with internal voids and delamination evident (Figure 8g).

The endotesta at the micropylar end was clearly divided into two layers, including the inner and outer layers, both of which were similar in density and were relatively denser than the conidial end. The outer endotesta was approximately 9.1 μm thick and fitted closely to the mesocarp without gaps. The inner endotesta was approximately 10.4 μm thick and fitted closely to the seed kernel, with gaps between the micropylar end and the seed embryo corresponding to the directly observed ellipsoidal structure, which provided a place for gas exchange for the growth of the seed embryo (Figure 8h).

#### 2.2.3. Scanning Electron Microscope Observation

To gain more details on the structure of the endotesta, a scanning electron microscope was employed to examine the structure of the three distinct segments within the endotesta of naturally mature Ginkgo seeds.

The observations revealed that the endocarp at the chalazal end exhibited two vascular traces corresponding to the inner part of the mesocarp. This part demonstrated a rough surface, characterized by more irregular protrusions of varying heights, which formed numerous furrowed interstices. In addition, it had a higher wax content than the endotesta at the micropylar end. This contributes to enhanced safeguarding of the seed kernel (Figure 9a). Conversely, the number of protrusions decreased in proximity to the mid-constriction line, resulting in a relatively smooth surface embellished with a longitudinal filiform ornamentation. Notably, the remnants of mesocarp-derived tubercles remained on this surface (Figure 9b).

The micropylar terminus of the endotesta featured an outer layer enveloped by a thin waxy and tuberculate layer, constituting the secondary protective layer of the seed embryo. This layer provides a certain degree of protection against external infestations. The epidermal microstructure exhibited a reticulate pattern characterized by a thick reticulate wall. Predominantly, this pattern consisted of elevated inconspicuous reticulate ridges that were meticulously demarcated and accompanied by shallow reticulate depressions and substantial variation in shape (Figure 9c). However, close to the middle constriction line, the reticulate structure vanished and the epidermal surface was smooth (Figure 9d).

The inner layer of the endotesta at the micropylar end had a distinct projection in the middle of the micropylar end, which was the structure wrapping and holding the embryo radicle (Figure 9e). Furthermore, this layer exhibited a waxy and tuberculate surface, constituting the initial safeguarding layer of the seed embryo. The reticulate wall formed the first depressional configuration, from the micropylar end to the middle constriction line. This depression was accentuated by a moderately prominent central projection that displayed an irregular shape. Notably, the region near the middle constriction line differed from the corresponding exterior region of the micropylar end. This distinction was characterized by a polygonal reticulation pattern alongside a delicate filamentous motif within the reticulum (Figure 9f).

### 2.3. Main Components of the Endotesta

#### 2.3.1. Basic Chemical Composition Analysis

Various fundamental components, including moisture, ash, protein, crude fat, and total soluble sugars within each part of the Ginkgo endotesta (Figure 10a), were determined using diverse analytical techniques.

#### 2.3.2. Analysis of Amino Acid Composition

A comprehensive spectrum of amino acids, comprising 7 essential amino acids (EAAs) and 10 non-essential amino acids (NEAAs), was identified within the endotesta of *Ginkgo biloba* (Appendix A). However, tryptophan remained undetected due to its degradation during hydrochloric acid hydrolysis. The total amino acid content (TAA), totaling 43.79 mg/g, encompassed the entire range of amino acids. Notably, glutamic acid (Glu) and aspartic acid (Asp) were present at substantial levels, while cysteine (Cys) and methionine (Met) were present at minimal concentrations, constituting only 1% of the TAA. The quantity of EAAs was 15.99 mg/g, yielding an EAA/TAA ratio of 36.52%. The NEAA content was 27.8 mg/g, resulting in an EAA/NEAA ratio of 57.51%, These ratios closely aligned with the FAO/WHO-recommended benchmarks for high-quality proteins (40% and 60%, respectively).

##### Flavor and Functional Amino Acids

In L-amino acids, a small side group generally elicits a sweet taste, whereas a large side group containing a basic functional group tends to result in a bitter taste. The presence of an aromatic group in the side group typically contributes to aromatic taste. Therefore, according to their taste characteristics, amino acids can be categorized into the following distinct taste classes: sweet amino acids (SwAAs), flavor amino acids (FAAs), bitter amino acids (BAAs), aromatic amino acids (ArAAs), and sour amino acids (SAAs). Specifically, the sweet amino acid group encompasses glycine, alanine, serine, threonine, and proline. The fresh amino acid group includes lysine, glutamic acid, and aspartic acid. The bitter amino acid group comprises valine, leucine, isoleucine, methionine, tyrosine, arginine, phenylalanine, and histidine. The aromatic amino acid group consists of phenylalanine, tyrosine, and cysteine. Lastly, the acidic amino acid group encompasses aspartic acid, glutamic acid, and histidine.

Amino acids can be divided into medicinal amino acids (MAAs) and branched chain amino acids (BcAAs) based on their functional characteristics. The medicinal amino acids include aspartic acid, glutamic acid, glycine, methionine, isoleucine, leucine, phenylalanine, tyrosine, and lysine. Meanwhile, the branched chain amino acids encompass valine, isoleucine, and leucine.

The endotesta of *Ginkgo biloba* exhibited the highest MAA content at 26.78 ± 0.51 mg/g, constituting 61.16% of TAAs (Figure 10b,c). The second-highest were the SwAAs at 14.96 ± 0.50 mg/g, representing 34.16% of the TAAs. Conversely, the ArAA content was the lowest at 3.02 ± 0.20 mg/g, accounting for only 6.9% of the TAAs. This observation underscores the health-related attributes of the endotesta of *Ginkgo biloba*, consistent with both its medicinal and nutritional qualities.

##### Nutritional Evaluation of Amino Acids

The amino acid score (AAS) can be determined through comparison with the WHO/FAO ideal protein model [23], using the essential amino acid content and their proportions within the endotesta. This approach facilitates visual assessment of the nutritional value of the endotesta. The first limiting amino acid (FLAA) is that with the minimum amino acid score, while a higher score close to 100% indicates closer alignment with the standard pattern. The chemistry score of amino acids (CSAA) denotes the ratio of essential amino acid content relative to the equivalent essential amino acids in the reference protein (ovalbumin), serving as an indicator of nutritional value. The essential amino acid index (EAAI) evaluates the similarity between the essential amino acid composition in the sample and that of the standard protein (ovalbumin), with a score close to 100 denoting a heightened nutritional quality. Additional parameters include the ratio of amino acid (RAA), the ratio coefficient of amino acid (RCAA), and the score of RCAA (SRCAA).

The results for AAS, CSAA, RAA and RCAA show that threonine achieved the highest AAS and CSAA, approaching 70, whereas methionine + cysteine had the lowest, indicating a severe deficiency in FLAA (Figure 10d). Plant proteins generally exhibit low levels of sulfur-containing amino acids, with several other amino acids scoring in the range of 30 to 50, which demonstrates an inadequate amino acid composition. The EAAI was calculated to be 30.85, which significantly differed from the amino acid profile of the standard protein, demonstrating the nutritional inadequacy of the protein. In addition, methionine + cysteine attained the lowest RAA and RCAA, indicating pronounced deficiency, while several other amino acids demonstrated ratio coefficients approaching 1 (Figure 10e). The SRCAA was quantified at 68, indicating the relatively balanced nutritional value of the endotesta, as evidenced by the amino acid ratio coefficient scores.

#### 2.3.3. Fatty Acid Composition and Content

A comprehensive analysis revealed the presence of 18 distinct fatty acids within the endotesta of *Ginkgo biloba* (Appendix A). Among these, five were unsaturated fatty acids (UFA), including palmitic, linoleic, transoleic, Dihomo-γ-linolenic and oleic acids, constituting a cumulative content of 4083.72 μg/g, accounting for 9.12% of the total fatty acids. Additionally, 13 saturated fatty acids were identified, with notable concentrations of 22-docosanoic acid (behenic acid), 24 eicosanoic acid, 28-eicosanoic acid, and 26-eicosanoic acid (>5000 μg/g). Remarkably, within the fatty acid spectrum, 22-docosanoic acid exhibited the highest abundance, surpassing that of other fatty acids by orders of magnitude. In contrast, 15-methyl-hexadecanoic acid was the least abundant compound.

#### 2.3.4. Analysis of Vitamin Content

Seven vitamins were identified in the endotesta of *Ginkgo biloba* (Appendix A). Notably, VC emerged as the predominant constituent, with a substantial concentration of 44.72 μg/g. This was followed by VC, VE and VB12, with contents spanning 17.22 to 16.17 μg/g. Meanwhile, VB1, VB2 and VB3 presented lower concentrations ranging from 7.26 to 4.66 μg/g, and VA exhibited the lowest content at 1.47 μg/g. The discernible vitamin composition within Ginkgo endotesta indicates its inherent nutritional value. This property suggests it as a viable natural resource for augmenting the vitamin content of various food products.

#### 2.3.5. Inorganic Element Content Analysis

In the endotesta of *Ginkgo biloba*, 17 inorganic elements were detected (Appendix A). These encompassed crucial macronutrients such as K, Ca, Mg, P, and Na, with K attaining the most prominent concentration at 11,195.59 mg/kg. Ca followed closely with a content of 3077.86 mg/kg, while Zn and S had contents of 1055.96 mg/kg and 901.53 mg/kg, respectively, Na was detected at 284.15 mg/kg. Furthermore, the endotesta exhibited an assortment of trace minerals essential for human physiology, including B, Ba, Co, Cr, Cu, Fe, Mn, Zn and S. Notably, the trace elements featured a significant diversity, with Co exhibiting the lowest concentration at 1.28 mg/kg.

#### 2.3.6. Nutritional Evaluation of Vitamins and Inorganic Elements

Utilizing the index of nutrition quality (INQ) approach, an assessment of five vitamins and eight inorganic elements within the endotesta was conducted (Figure 10f). The INQ for Na was less than 1, indicating insufficiency. Conversely, all other indicators exhibited INQ values exceeding 2, with K, Cu, Mg, Fe, and Mn displaying INQ values within the range from 10 to 25. These results demonstrate the potential of the inner seed coat as a valuable resource for these nutrients, thereby offering a viable avenue to supplement the diets of specific populations.

The nutritional quality index of VC remained below 2, satisfying the requisite intake levels for human physiological needs. Conversely, the nutritional quality index of VB12, VB1, and VB2 surpassed 15, establishing the endotesta of *Ginkgo biloba* as a substantial source of compounds suitable for addressing the dietary requirements of distinct population subsets.

#### 2.3.7. Correlation Analysis of Nutrient Composition

To investigate the relationships among the four nutrients in the endotesta of *Ginkgo biloba*, a correlation analysis was conducted on the nutritional assessment index (Figure 11). The results revealed a notable positive correlation among the essential amino acids (*p* < 0.05), an absence of correlation between the fatty acids, a significant negative correlation among vitamins (*p* < 0.05), and an absence of correlation among inorganic elements. Remarkably, the essential amino acids exhibited positive correlations with VE and Fe (*p* < 0.05), whereas VC and VB1 were negatively correlated with inorganic elements (*p* < 0.05). This highlights the reciprocal contributions of VE and Fe elements, as well as VC and VB1, alongside inorganic elements, in addressing human nutritional requirements.

## 3. Discussion

### 3.1. Developmental Origin of the Endotesta

Extensive investigations into the genesis and development of the Ginkgo seed coat have yielded significant insights into the occurrence and developmental processes of both the sarcotesta and sclerotesta [24,25,26]. In contrast, the origin and formation of the endotesta remain less definitively elucidated [15].

In June, the cell walls of most independent nuclei within the female gametophyte of *Ginkgo biloba* were formed. This process entailed encircling the female gametophyte with the developing cell wall. Subsequently, the macrospore membrane underwent gradual disintegration and degradation. This transformation was facilitated by the growth of primordial neck cell divisions within the female gametophyte, the emergence of tent columns, the formation of bead core tissue enveloping tent columns, and interactions with the macrospore membrane [27,28,29,30,31].

In this study, we employed semi-thin and ultrathin sections to meticulously examine the progression of endotesta development. Our observations revealed that the endotesta of *Ginkgo biloba* mainly originated from the inner bead peridium, with a portion of the bead core tissue. In addition, the macrospore membrane played a noteworthy role in shaping the morphology of the endotesta. This discovery challenges the previous assumption that the endotesta could arise solely from the inner bead peridium or exclusively through macrospore membrane development [15]. On the other hand, the adjacent bead core tissue contributed to the formation of the inner endotesta, deviating from the conventional concept that its origin is the bead peridium tissue. Consequently, from a developmental anatomical standpoint, this structure may not entirely qualify as the endotesta. Our study further revealed the presence of a macrospore membrane between the residual peridium and the female gametophyte. Moreover, the cell wall remnants of the inner peridium and degraded peridium were closely integrated with the macrospore membrane.

### 3.2. Structural Characteristics of the Endotesta

In previous descriptions, the endotesta of *Ginkgo biloba* have often been described as membranous. However, other scholars have provided more detailed descriptions, characterizing the upper endotesta as papery and possessing an opaque, greyish-white appearance. In addition, the lower endotesta exhibits a dual-layer structure, with one layer adjoining the sarcotesta and the other attached to the seed kernel. Both layers manifest as translucent, golden-brown membranous structures [15,32,33].

In this study, we conducted a comprehensive and systematic examination of the apparent morphology and internal structure of the endotesta. This analysis included direct observations, micro-CT imaging and SEM. Our findings revealed distinctive characteristics within the endotesta of the mature *Ginkgo biloba* seed. The endotesta emerged as a thin and dry layer exhibiting different morphological attributes above and below the mid-constriction line. Specifically, within the area of the mid-constriction line region, a “herringbone” pattern prevailed, with the upper part presenting a monolayer structure. This segment was dark brown or greyish-white and featured two vascular bundle traces at the chalazal end. Its surface texture was rugged, covered with a substantial waxy layer serving to protect the embryo of the seed. The outermost regions of the endotesta demonstrated a loose texture characterized by internal gaps and a papery surface. The lower endotesta was divided into two distinct layers, one inside and the other outside, formed by the beads and bead cores. Despite their differing origins, both layers exhibited similar morphologies and structures. The outer endotesta appeared translucent and russet, closely resembling the sarcotesta, with minimal gaps. It was covered with a thin waxy and tumorous layer, characterized by a dense membranous structure. Conversely, the inner endotesta layer with its russet and translucent characteristics, adhered closely to the seed kernel. The micropylar end featured a milky-white ellipsoidal structure, with a cavity between it and the radicle of the seed embryo. A thin waxy and verrucose layer on its surface was dense and membranous. A substantial interlayer cavity facilitated outward embryo elongation.

This intricate micropylar endotesta structure could be used for water retention, supported by surface wax and tumor layers that are impermeable to water, thereby restraining transpiration losses. Ginkgo seeds, categorized as “recalcitrant”, are susceptible to dehydration [34]. The structural characteristics of the endotesta are crucial in ensuring water retention, equipping *Ginkgo biloba* to survive arid environments. As an ancient relict plant, the survival of *Ginkgo biloba* has necessitated evolutionary adaptation to changing environments balanced with the preservation of ancestral traits [33,35,36].

Therefore, distinctive attributes, such as the dual-layered structure at the micropylar end of the endotesta and the presence of a waxy tumor layer on the surface of the endotesta, indicate that Ginkgo is characterized by structures adapted to dry conditions. In addition, during the endotesta removal, it was found that the seed kernel without its enveloping endotesta exhibited heightened susceptibility to desiccation and consequent hardening under identical indoor conditions. Subsequent exposure to humidity led to mold proliferation and decay, substantiating the positive role of the Ginkgo endotesta in forestalling dehydration, microbial invasion, and oxidative inactivation of the seed kernel. In light of these findings, strategic regulation of the developmental process of the endotesta and elevation of surface wax content can facilitate promising avenues for augmenting the storage longevity and preservation efficacy of ginkgo seeds [33,37,38].

### 3.3. Main Components of the Endotesta

There has been limited exploration of the compositional aspects of the *Ginkgo biloba* endotesta. In this study, we conducted a comprehensive assessment of its fundamental constituents, including amino acids, fatty acids, vitamins and inorganic elements. The endotesta of *Ginkgo biloba* had a water content of 37.40%, protein content of 9.94%, fat content of 3.35%, soluble sugar content of 4.09%, and ash content of 2.33%. Notably, the fat and ash contents closely approximated those of the seed kernel, while the protein and soluble sugar contents exhibited a marginal decrease in comparison with the seed kernel [39].

A comprehensive array of constituents was identified in the endotesta of *Ginkgo biloba*, including 17 amino acids, 18 fatty acids, 7 vitamins, and 15 inorganic elements. The TAA content was 43.79 mg/g. EAAs comprised 36.52% of the TAA content, and EAA/NEAA was 57.51%, closely approximating the ideal model of high-quality protein recommended by the FAO/WHO [23]. Among the fatty acids, behenic acid was predominant. The UFAs mainly included linoleic acid, oleic acid, and dihomo-γ -linolenic acid, albeit with a lower content than that of the seed kernel [40]. The vitamin contents of the endotesta, particularly VE, VB1, VB2, and VC, surpassed those of the seed kernel by factors of 1.6 to 12.9. Moreover, both VE and VC exerted a significant antioxidant effect, whereas VB1 and VB2, essential vitamins, intricately participated in bodily metabolism playing a pivotal role in sustaining the nervous, digestive, and visual function systems [41,42]. Consequently, the Ginkgo endotesta served as a viable raw material for natural additive development aimed at enhancing food vitamin content. The Ginkgo endotesta demonstrated an abundance of diverse inorganic elements, characterized by a “K-rich and P-poor” profile. Remarkably, its Ca content surpassed that of the seed kernel by approximately eight-fold, while the Mg content exceeded that of the sclerotesta by a factor of four [43]. K plays a pivotal role as an electrolyte in the human physiology. Ca regulates numerous bodily systems. Mg is integral for maintaining neural excitability and supporting normal bone growth [44]. Despite the limited exploration of the biological constitution of *Ginkgo biloba* endotesta, systematic investigation into variations attributed to origins, varieties, and cultivation conditions holds immense promise. The accumulation of comprehensive data would provide detailed insights into the biological characteristics of the endotesta, thereby serving as a robust reference for optimizing the medicinal and edible potential of *Ginkgo biloba* endotesta.

## 4. Materials and Methods

### 4.1. Specimen Collection

The experimental materials in this study consisted of a mature female *Ginkgo biloba* plant, approximately 20 years old, cultivated on the campus of Nanjing Forestry University (Nanjing, China). Sampling commenced with the emergence of pollination drops in mid-April and extended to late September (9.25–9.30) when seed ripening occurred (Figure 1). Throughout this timeframe, samples were collected every 5 d. During each sampling event. 30 specimens exhibiting normal development were collected randomly selected and divided equally into two groups, each containing 15 specimens. One group underwent solid dissection to observe and document developmental progression, and images were captured for documentation. The second group underwent tissue sectioning involving dissection of the endotesta to obtain 2 mm × 2 mm × 1 mm sections. These sections were then treated with 2.5% glutaraldehyde fixative and stored at 4 °C in a refrigerator. Subsequently, 30 naturally mature Ginkgo seeds were selected. The outer and middle bony seed coats were removed and the endotesta was extracted using forceps. The extracted endotesta components were also treated with 2.5% glutaraldehyde fixative and stored at 4 °C in the refrigerator.

### 4.2. Transmission Electron Microscopy

Ginkgo seeds representing distinct developmental phases were subjected to TEM analysis following fixation were fixed in 2% glutaraldehyde in 0.1 M phosphate buffer for 24 h. Subsequent steps involved washing the samples in 0.1 M phosphate buffer (3 × 15 min) and post-fixing in 1% OsO4 buffer for 2 h, all performed at pH 7.2 and at 25 °C. The dehydration procedures included sequential immersions in graded ethanol concentrations (30%, 50%, 75%, 80%, and 95%) for 15 min each, followed by three rounds of 100% ethanol treatment for 15 min each. Additional dehydration involved the transition solvent propylene oxide (PO) (3 × 15 min) and a mixture of PO: Epon in varying ratios (1:1 for 3 h and 1:2 for 3 h). Subsequently, the specimens were infiltrated with Epon for 2 d at room temperature, followed by embedding and polymerization in fresh Epon at 30 °C for 24 h, 45 °C for 12 h and 60 °C for 24 h. Semi-thin sections with a thickness of 1 μm were obtained for microscopic analysis after staining with 0.5–1% Toluidine Blue (TBO), facilitating microstructural observations through light microscopy. Ultrathin sections (70 nm) were cut using glass knives on a Leica UC 6 ultramicrotome. These sections were then post-stained using saturated uranyl acetate and 1% lead citrate for 10 min, followed by air drying. TEM imaging was conducted using a 120 kv JEM-1400 device (JEOL Ltd., Beijing, China).

### 4.3. Scanning Electron Microscopy

The four segments of the endotesta, encompassing the outer endotesta at the bead hole end, inner endotesta at the bead hole end, endotesta at the coaptation end and the mid-constriction area, were submerged in a 70% ethanol solution containing 0.5% Triton X-100. These segments were refrigerated overnight at 4 °C. On the subsequent day, the endotesta were subjected to 5 s of sonication for cleansing and rinsed with water. A graded ethanol series of 80%, 90%, and 95% (each for 15 min) was applied for dehydration, followed by two rounds of complete dehydration using 100% ethanol (each for 15 min). After critical point drying (ES-2030, Hitachi, Tokyo, Japan), the treated endotesta samples were affixed to a SEM sample holder on opposite sides (left and right) through the application of double-sided adhesive tape. These samples were sputter-coated with gold before being examined using a Quanta 200 (FEI, San Jose, CA, USA) scanning electron microscope, operated at 25 kV.

### 4.4. Micro-Computed Tomography

Naturally mature Ginkgo seeds sheltered from direct light, underwent X-ray 3D micro-CT scanning at 20 °C and 65% relative humidity using NanoVoxel-4000 equipment (Sanying Precision Instruments Co., Ltd., Tianjin, China).

### 4.5. Determination of the Main Components in the Endotesta

The endotesta were carefully removed using forceps, followed by freezing, drying, crushing and sieving. The resulting material was stored at −80 °C in a refrigerator until the determination of parameters, including crude protein, crude fat, soluble sugars, inorganic elements, amino acids, and fatty acids. Freshly extracted endotesta samples were used to ascertain the moisture and vitamin content. These samples were rinsed in distilled water, air-dried under ambient conditions, shielded from light, and then crushed before direct determination.

### 4.6. Methodology for Assessing Nutritional Value of Amino Acids

The proportions of essential amino acids (EAAa) and non-essential amino acids (NEAAs) relative to the total amino acids (TAAs) within *Ginkgo biloba* endotesta were calculated and compared with the revised 2013 model spectrum of essential amino acid contents by the World Health Organization and the Food and Agriculture Organization of the United Nations (FAO/WHO) [23]. Employing this spectrum, several indices and scores were determined, such as the amino acid score (AAS), chemistry score of amino acids (CSAA), essential amino acid index (EAAI), ratio of amino acids (RAA), ratio coefficient of amino acids (RCAA), and the score of ratio coefficient of amino acids (SRCAA), as detailed in Equations (1)–(6):(1)$$AAS=\frac{a}{A\left(\frac{FAO}{WHO}\right)}\times 100$$
(2)$$CSAA=\frac{a}{A\left(egg\right)}\times 100$$
(3)$$EAAI=100\times \sqrt[{7}]{\frac{Ile}{Ile\left(\frac{FAO}{WHO}\right)}\times \frac{Leu}{Leu\left(\frac{FAO}{WHO}\right)}\times \frac{Lys}{Lys\left(\frac{FAO}{WHO}\right)}\times \frac{Thr}{Thr\left(\frac{FAO}{WHO}\right)}\times \frac{Val}{Val\left(\frac{FAO}{WHO}\right)}\times \frac{Met+Cys}{Met+Cys\left(\frac{FAO}{WHO}\right)}\times \frac{Phe+Tyr}{Phe+Tyr\left(\frac{FAO}{WHO}\right)}}$$
(4)$$RAA=\frac{a}{A\left(\frac{FAO}{WHO}\right)}$$
(5)$$RCAA=RAA/\bar{RAA}$$
(6)$$SRCAA=100-100\times \sqrt{\frac{\sum {\left(RCAA-\bar{RCAA}\right)}^{2}}{7}}$$

In Equations (1)–(6), a denotes the content of an EAA in the endotesta; $A\left(\frac{FAO}{WHO}\right)$ denotes the content of the same amino acid in the FWO/WHO scoring standard model; $A\left(egg\right)$ denotes the content of the same amino acid in the egg white protein; $\bar{RAA}$ denotes the mean value of RAA; and $\bar{RCAA}$ denotes the mean value of RCAA.

Typically, the values of AAS, CSAA, EAAI, and SRCAA (nearing 100) indicate a better balance, enhanced nutritional value, and heightened nutrient content in the sample. Similarly, RAA and RCAA values approaching 1, signify greater alignment with the FAO/WHO essential amino acid recommendations.

### 4.7. Methodology for the Evaluation of Vitamins and Mineral Elements

The index of nutrition quality (INQ) is an indicator of the nutritional quality of food, assessing both the degree to which each nutrient in the food fulfills bodily requirements and its ability to satisfy caloric energy needs. The INQ was applied to evaluate the nutritional content of five vitamins and eight mineral elements within Ginkgo endocarps, using the equation presented in (7):(7)$$INQ=\frac{I/I_{n}}{E/E_{n}}$$
where I denotes the content of a certain nutrient; $I_{n}$ denotes the recommended intake of that nutrient; E denotes the energy provided by the sample; and $E_{n}$ denotes the recommended energy intake. An INQ < 1, signifies a nutrient content lower than the recommended intake, potentially leading to inadequate long-term consumption and diminished nutritional quality. Conversely, an INQ value equal to or greater than 1 indicates a nutrient content that matches or surpasses the recommended intake, reflecting acceptable nutritional quality. An INQ value exceeding 2 denotes the food as a substantial source of the specific nutrient.

### 4.8. Data Processing

The data were processed using a suite of software tools including Photoshop CS 5, Origin 2023, IBM SPSS Statistics 27, and Excel 2021.

## 5. Conclusions

In this study, we conducted a comprehensive and systematic analysis of the developmental process of the Ginkgo endotesta. Distinct segments of the endodermis were sampled separately, and semi-thin and ultrathin sections were employed to observe the cellular morphological attributes and internal ultrastructural composition. This approach has facilitated the elucidation of the developmental process and ontogenetic origins of the endotesta. To further comprehend the external architecture of the mature Ginkgo endotesta, SEM and micro-CT were utilized, revealing the presence of two vascular bundles at the chalazal end. In addition, an assessment of the nutrient composition and nutritional significance of *Ginkgo biloba* endotesta was conducted and aligned with the FAO/WHO recommendations. This evaluation provides a pivotal reference for optimizing the medicinal and edible potential inherent to *Ginkgo biloba* endotesta. This study proposes a robust theoretical foundation for the comprehensive exploration and effective utilization of the attributes and resources of the Ginkgo endotesta.

## Figures and Tables

**Figure 1 plants-12-03560-f001:**
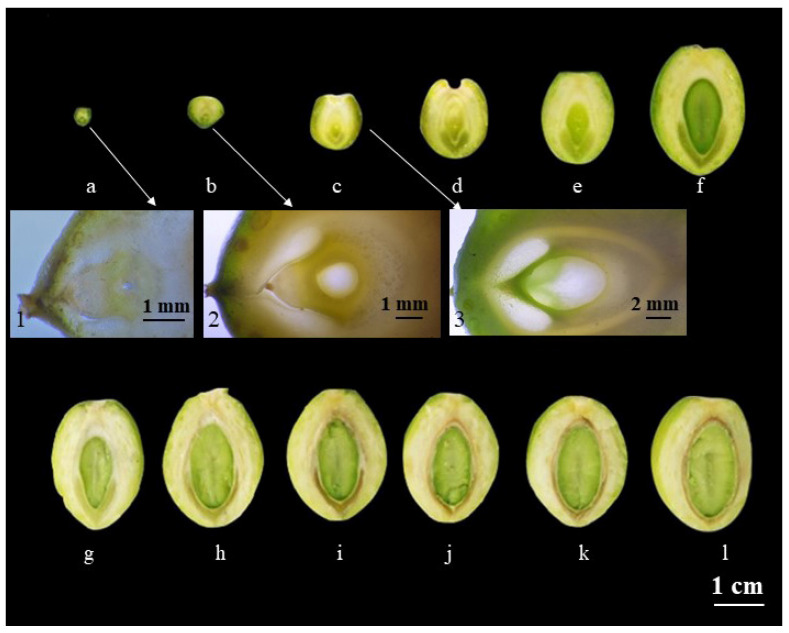
Developmental changes in the endotesta of Ginkgo ovules (mid-April to early July). (**a**,**b**) April; (**c**–**f**) May; (**g**–**k**) June; (**l**) July. Numerals 1, 2, and 3 denote the ovules observed under stereomicroscopy during corresponding periods (**a**–**c**).

**Figure 2 plants-12-03560-f002:**
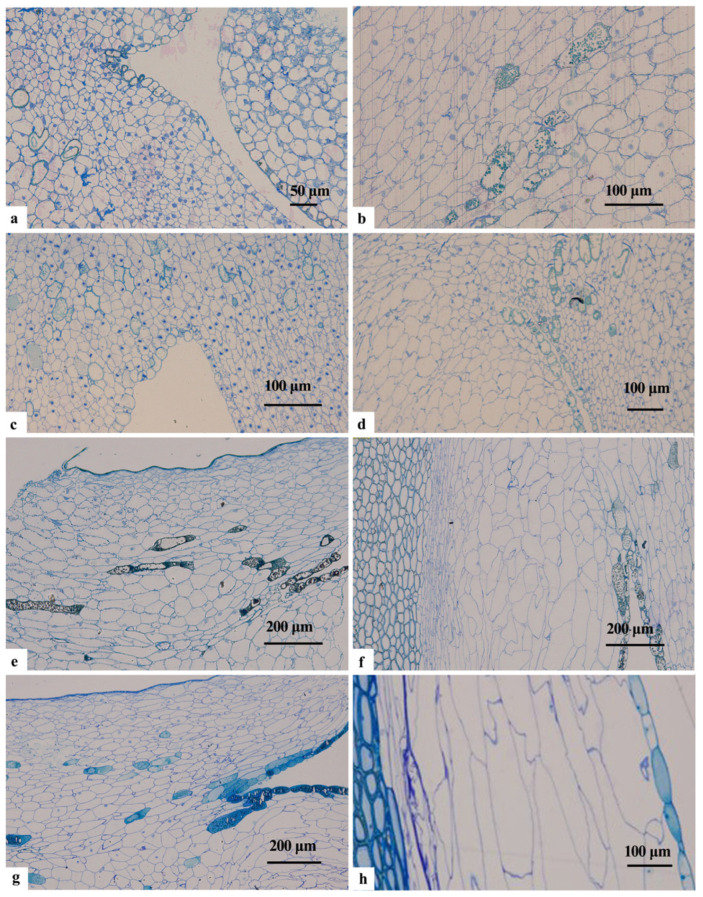
Microstructure of the inner integument of *Ginkgo biloba* from April to June. (**a**,**b**) Inner integument in April: (**a**) micropylar end; (**b**) chalazal end. (**c**,**d**) Micropylar end of inner integument in May. (**e**–**h**) Inner integument in June: (**f**,**g**) micropylar end; (**e**,**h**) chalazal end. CS: Conoid structure; II: Inner integument; MM: Megaspore membrane; Nu: Nucellus; OI: Outer integument.

**Figure 3 plants-12-03560-f003:**
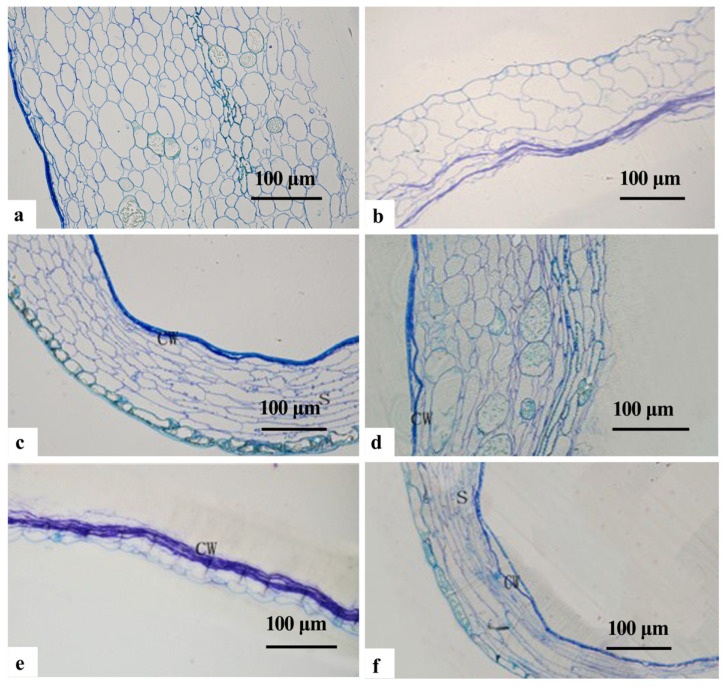
Microstructure of the inner integument of *Ginkgo biloba* in July. (**a**,**c**,**e**) Inner integument at the chalazal end. (**b**,**d**) Inner integument at the micropylar end. (**f**) Inner nucellar tissue. CS: Conoid structure; CW: Cell wall; II: Inner integument; MM: Megaspore membrane; Nu: Nucellus; OI: Outer integument; S: Starch granules.

**Figure 4 plants-12-03560-f004:**
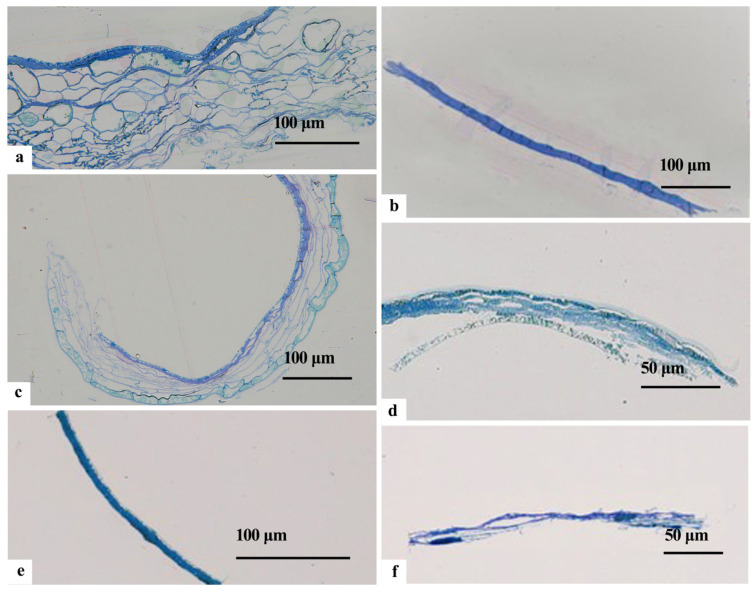
Microstructure of the inner integument of *Ginkgo biloba* in August and September. (**a**,**f**) Inner integument at the chalazal end. (**b**,**d**,**e**) Inner integument at the micropylar end. (**c**) Inner nucellar tissue.

**Figure 5 plants-12-03560-f005:**
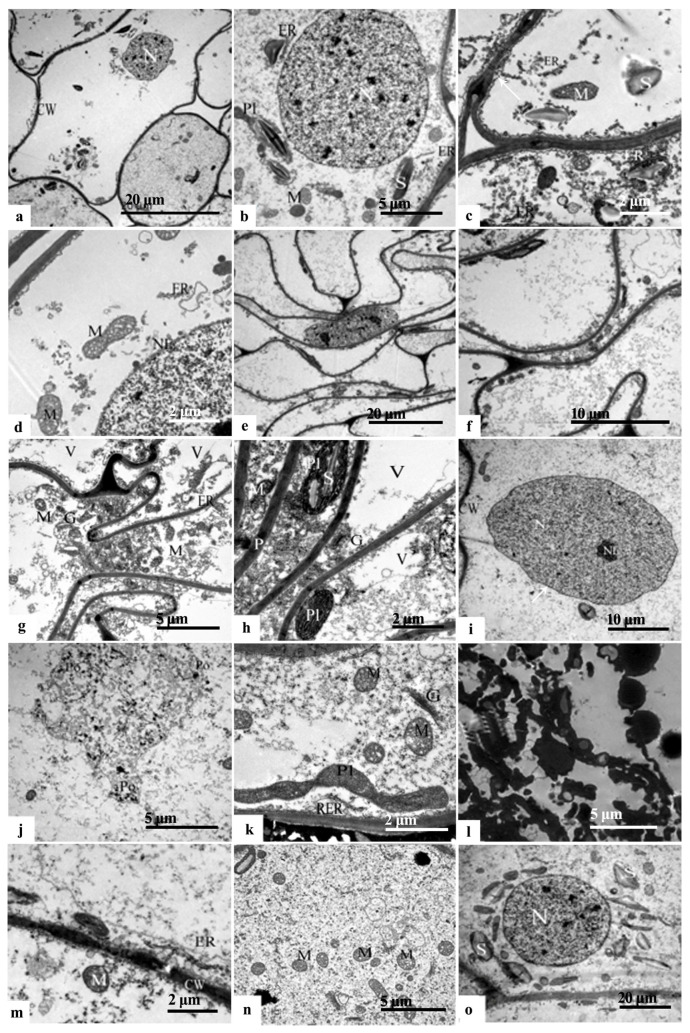
TEM photomicrographs of the endotesta in May in *Ginkgo biloba*. (**a**–**d**) Inner integument at the chalazal end. (**e**–**h**) The outer endotesta at the micropylar end; the arrows show the intercellular filament in (**c**,**g**). (**i**–**o**) Inner endotesta at the micropylar end; the arrow shows the cell membrane in (**i**,**j**) which show protein substances around the polyribosome. CW: Cell wall; ER: Endoplasmic reticulum; G: Golgiosome; M: Mitochondria; N: Nucleus; P: Plasmodesmus; Pl: Plastid; Po: polyribosome; RER: Rough endoplasmic reticulum; S: Starch granules; V: Vacuoles.

**Figure 6 plants-12-03560-f006:**
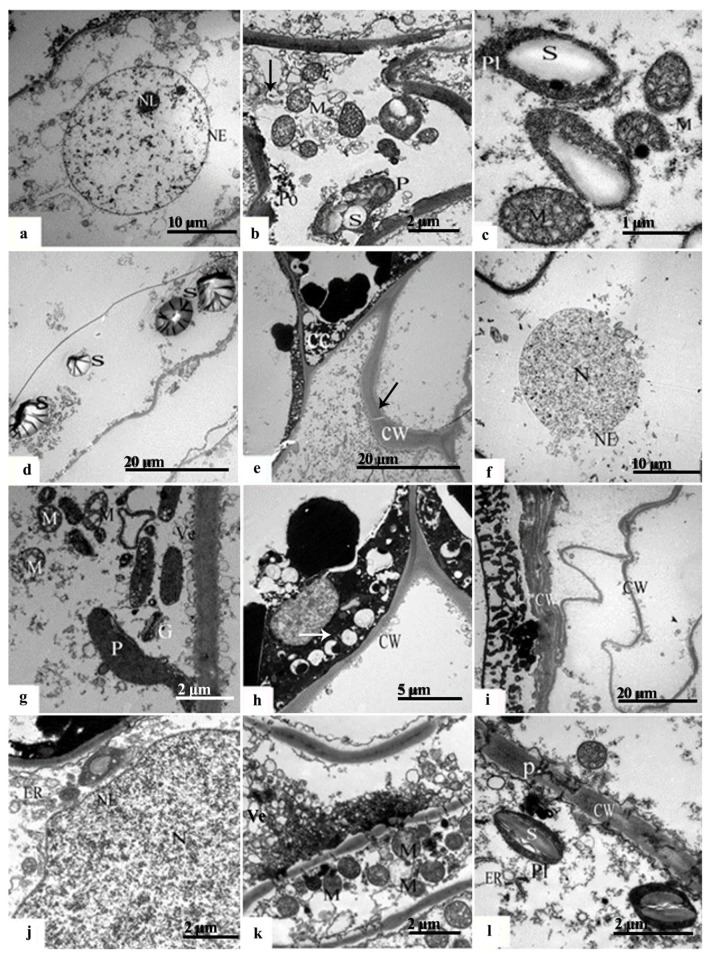
TEM photomicrographs of the endotesta in June in *Ginkgo biloba*. (**a**–**d**) Inner integument at the chalazal end, and the black arrow in (**b**) shows inclusions, which contains protein, fatty acid and other starch granules. (**e**–**h**) Outer endotesta at the micropylar end; the black arrow in (**e**) shows thickened cell walls, and in (**h**), the white arrow shows inclusions. (**i**–**l**) Inner endotesta at the micropylar end; the (**k**) shows the intercellular filament. CW: Cell wall; CC: Cell Corner; ER: Endoplasmic reticulum; G: Golgiosome; M: Mitochondria; N: Nucleus; NE: Nuclear membrane; P: Plasmodesmus; Po: Polyribosome; Pl: Plastid; RER: Rough endoplasmic reticulum; S: Starch granules; V: Vacuoles; Ve: Vesicle.

**Figure 7 plants-12-03560-f007:**
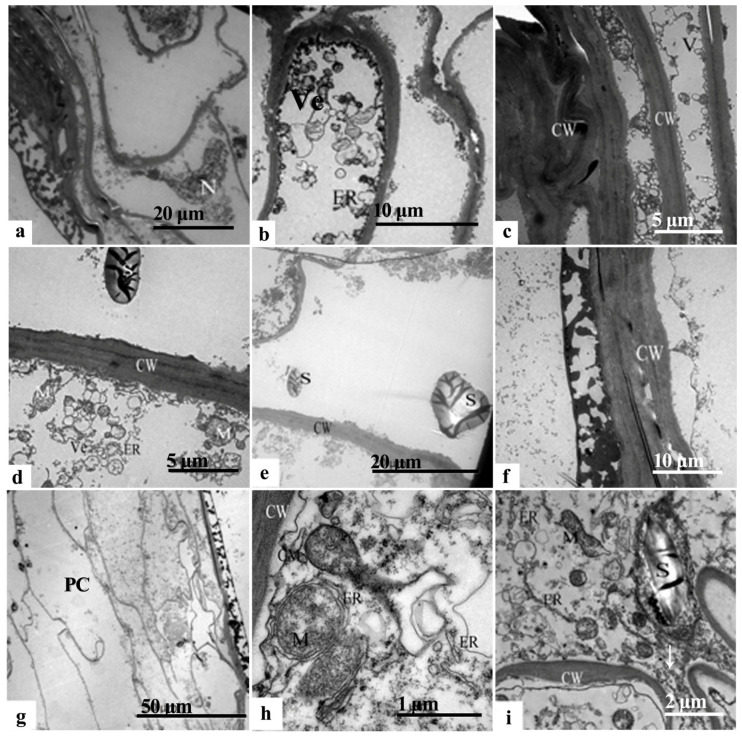
TEM photomicrographs of the endotesta in July in *Ginkgo biloba*. (**a**–**d**) Inner integumen at the chalazal end. (**e**–**i**) Inner endotesta at the micropylar end; the white arrow in (**i**) shows inclusions. CW: Cell wall; ER: Endoplasmic reticulum; M: Mitochondria; N: Nucleus; NE: Nuclear membrane; PC: Periplasmic cells; S: Starch granules; V: Vacuoles.

**Figure 8 plants-12-03560-f008:**
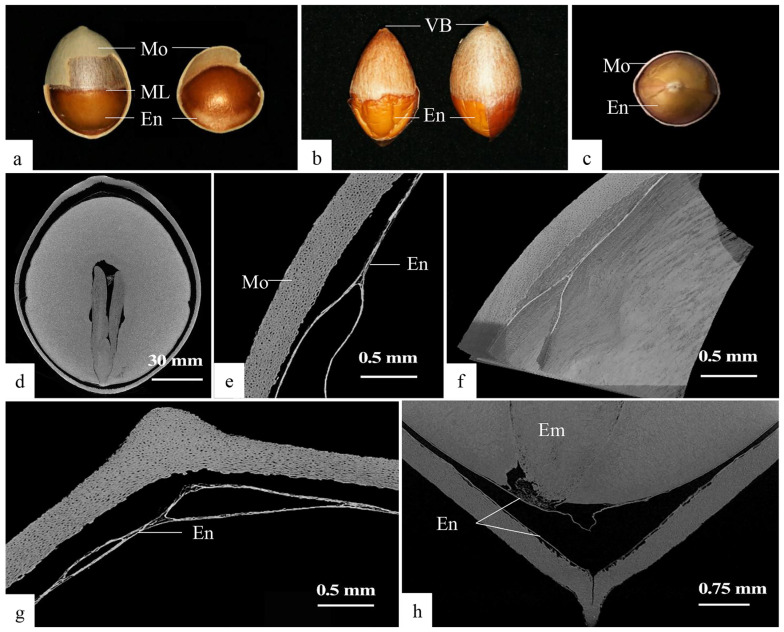
Apparent structure and CT imaging of the endotesta. (**a**) Middle line and upper and lower morphological structures in the endotesta. (**b**) Traces of vascular bundles. (**c**) Micropylar end structures of the endotesta. (**d**) Overall structure of Ginkgo seeds with episperm removed. (**e**) ‘Herringbone’ shaped endotesta. (**f**) Three-dimensional view of the ‘herringbone’ structure. (**g**) Endotesta at end of chalazal end. (**h**) Endotesta at end of micropylar end. En: Endotesta; Em: Embryo; ML: Middle line; Mo: Mososperm; VB: Vascular bundle.

**Figure 9 plants-12-03560-f009:**
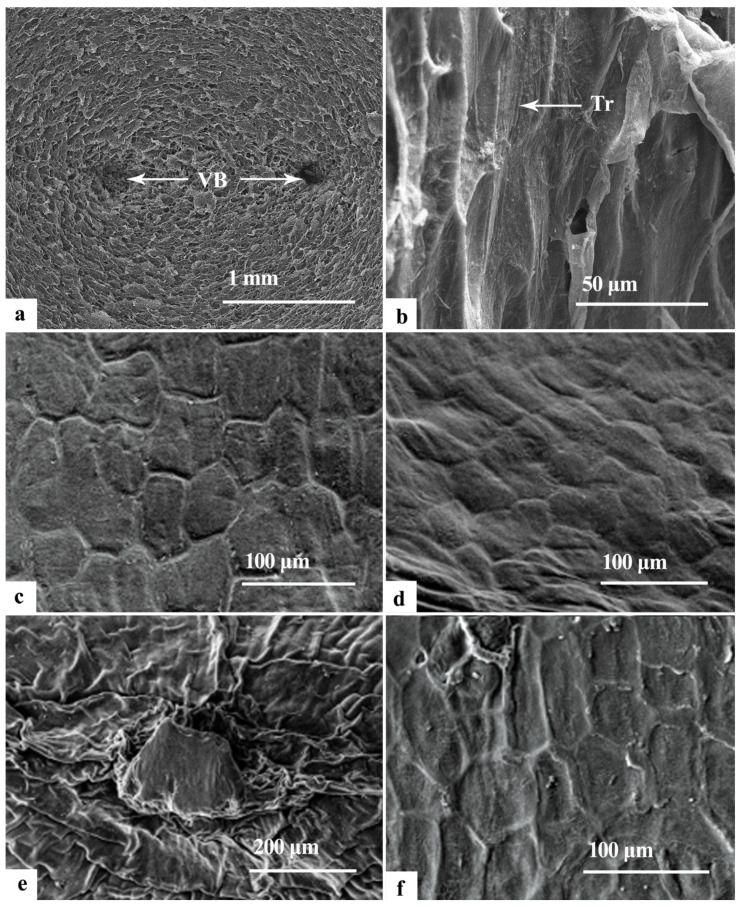
SEM photomicrographs of the endotesta in *Ginkgo biloba*. (**a**,**b**) Endotesta at the chalazal end. (**c**,**d**) Outer endotesta at the micropylar end. (**e**,**f**) Inner endotesta at the micropylar end. Tr: Tracheid; VB: Vascular bundle.

**Figure 10 plants-12-03560-f010:**
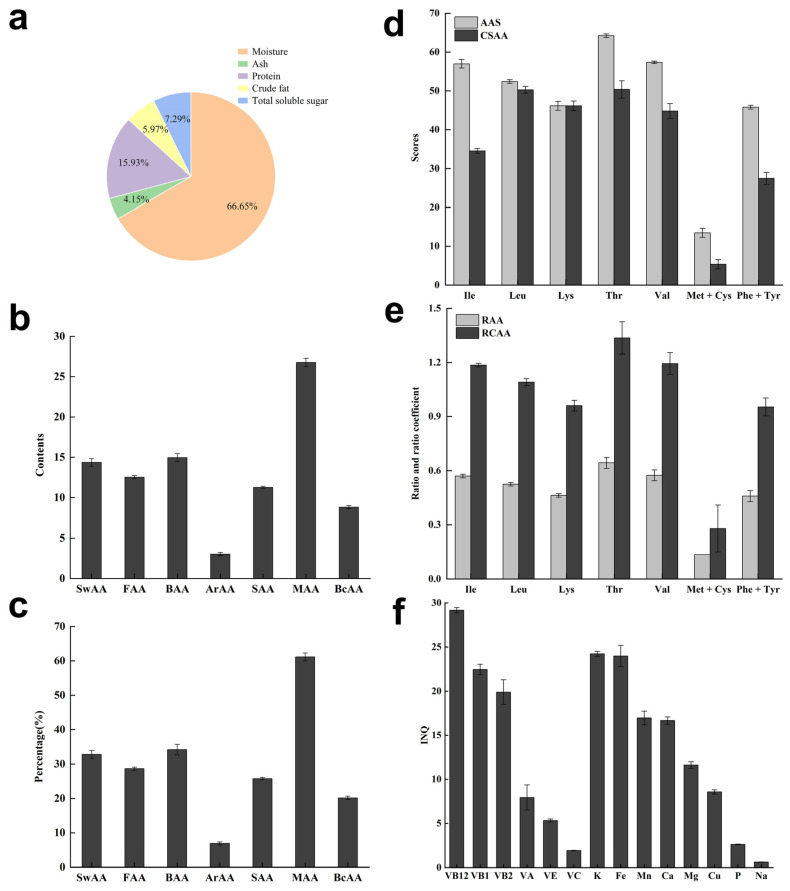
Basic chemical compositions and nutritional value evaluation in the endotesta of *Ginkgo biloba*. (**a**) Basic chemical compositions. (**b**) Contents of flavor and functional amino acids. (**c**) Percentage of TAA of flavor and functional amino acids. (**d**) Essential amino acids scores and chemical scores (AAS and CSAA). (**e**) Ratios and ratio coefficients of amino acids (RAA and RCAA). (**f**) INQ values for eight inorganic elements and six vitamins. Note: Except for the moisture index, all other indexes are calculated on a dry basis.

**Figure 11 plants-12-03560-f011:**
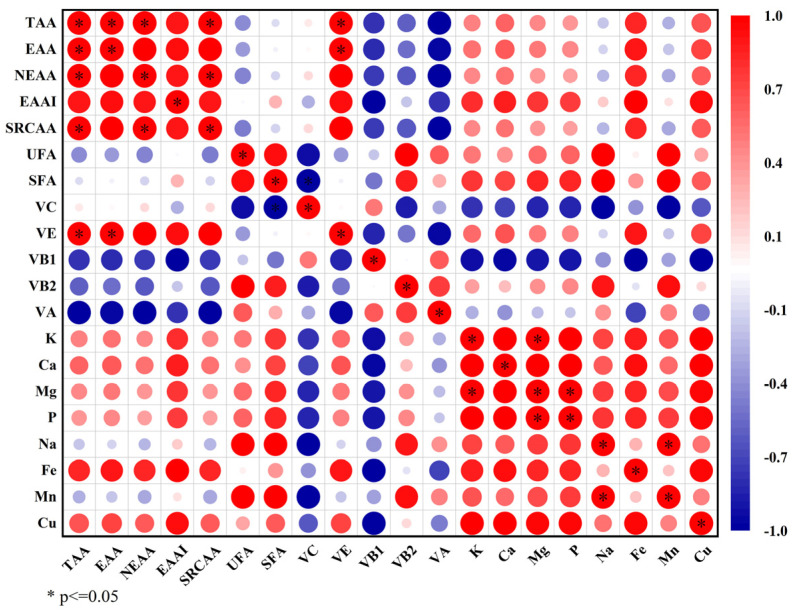
Correlation analysis of the four nutrient components. The heat map was created according to the result of Spearman’s correlation analysis of four nutrient components. The protein data are represented by TAA, EAA, NEAA, EAAI and SRCAA; the UFA and SFA represent fatty acids; the vitamins included are VC, VE, VB1, VB2 and VA; and the inorganic elements are K, Ca, Mg, P, Na, Fe, Mn, Cu.

## Data Availability

The article contains all the information required to support its conclusions.

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
