# Peer review of "Morphological Characteristics, Ultrastructure, and Chemical Constituents of the Endotesta in Ginkgo (Ginkgo biloba L.)"

_plants, 2023, doi:10.3390/plants12203560_

Round 1
Reviewer 1 Report
(1) The manuscript is discuss about the chemical, ultrastructure, and morphological components of the endotesta in Ginkgo (Ginkgo biloba L.) (2) The article in Original and I have not found significant similarity index between this article and other articles. (3) The results showed that the endotesta had a membranous structure primarily come from the inner bead peridium, segment of bead core tissue, and macrospore membrane. (4) The manuscript has written very well, its just need Minor revision in Title, Abstract and arranging keywords on the basis of alphabetic order. Also references format should be double-checked. (5) The conclusion has written very well, and it dose not need any changes. (6) The references appropriate, enough and updated, but format and DOI should be considered and re-corrected. (7) Tables and figure are clear and they have good quality, and they do not need any changes.(8) Please add English name of the plant before its scientific name.
(9) The abstract has written very well, but authors should arrange and write keywords on the basis of alphabetic orders, and it is recommended to do not use those keywords which have been used in the Title of the manuscript.
(10) The results and discussion has written very well.
(11) Conclusion is well-written and it does not need any changes.
(12) Authors need to add DOI for all references.
(13) Please, check references 12 and 13, it seems they are NOT correct.
The articles has written very well, and it just need Minor revision.
Author Response
Response to reviewer
Dear Editor and Reviewer:
Thank you for carefully reading our manuscript and offering us an opportunity to improve our article: Morphological, Ultrastructure, and Chemical Constituents of the Endotesta in Ginkgo biloba L. (plants-2629358). We appreciated the reviewer’s insightful comments and constructive suggestions very much. All these comments/suggestions were highly valued and addressed in this revision. We hope the revised manuscript has now met the standard of the journal. As required, all revisions have been marked using ‘track changes’ so that you can see them in MS Word.
Comment 1: The manuscript has written very well, its just need Minor revision in Title, Abstract and arranging keywords on the basis of alphabetic order. Also references format should be double-checked.
Response 1: Thank you for advising, we have revised the Title to (Morphological, Ultrastructure, and Chemical Constituents of the Endotesta in Ginkgo (Ginkgo biloba L.)), Rearranging keywords on the basis of alphabetic order (chemical constituents; endotesta; Ginkgo biloba L.; ultrastructure). And references format has been checked.
Comment 2: The references appropriate, enough and updated, but format and DOI should be considered and re-corrected.
Response 2: The references format and DOI have been added.
Comment 3: Please add English name of the plant before its scientific name.
Response 3: The English name of the plant (Ginkgo) have added before its scientific name. Morphological, Ultrastructure, and Chemical Constituents of the Endotesta in Ginkgo (Ginkgo biloba L.)
Comment 4: The abstract has written very well, but authors should arrange and write keywords on the basis of alphabetic orders, and it is recommended to do not use those keywords which have been used in the Title of the manuscript.
Response 4: Thanks for your suggestion. The keywords have been changed as follows: chemical composition; development; Ginkgo endotesta; membranous structure.
Comment 5: Authors need to add DOI for all references.
Response 5: Thank you for your reminding. We have added DOI for all references.
Comment 6: Please, check references 12 and 13, it seems they are NOT correct.
Response 6: We have checked and corrected the references 12 and 13.

Reviewer 2 Report
The manuscript Morphological, ultrastructure, and chemical constituents of the endotesta in Ginkgo biloba L. by Fangdi Li, Ganping Liu, Linying Zhao, Xiaoge Gao, Zhuolong Shen, Fuliang Cao, Qirong Guo presents a detailed comprehensive study of the seed coat and seeds of ginkgo from the point of view of morphology , composition and prospects for use and storage.
The value of the manuscript in the traditional interest in this type of ancient flora is accompanied by a number of new data obtained as a result of the study.
The manuscript contains all the necessary parts and is formatted according to the rules.
There are a number of comments that need to be addressed before its publication.
Significant errors:
The monthly development scheme should be increased and should contain the time range in numbers, since the scheme should also be applicable to researchers in other regions located in other zones and different climates (this is an international journal).
At the end of the introduction, the authors summarize the results of the work, rather than formulating problems and hypotheses to solve or test which the study was aimed at. This paragraph is suitable for a conclusion or discussion and should be reformulated.
Figures 5,6,7 should be significantly enlarged and the signatures and the reflection of abbreviations in the signatures and photographs should be checked. They don't match!
Layers of fabrics must be clearly marked - with color, arrows, etc. Such works are valuable for use and must be done with high quality and not contain discrepancies. The quality of the images should be improved.
Significant morphological deposits, given that the authors paid attention to the composition, should be designated and, if they were not identified, discussed. For example, such deposits are usually characterized as deposits of phenolic compounds. ...Thickening of the cell wall in this layer of cells may be associated with..
Discussion of functional versions of morphological features is required.
Figures 8 and 9 do not contain symbols.
Figure 10 - very small, no statistics in the caption.
The caption of Figure 11 must contain complete information, since the images are viewed separately from the article and must be self-sufficient (object, method, processing method, etc.)
The work is generally good and can be published after editing
Author Response
Response to reviewer
Dear Editor and Reviewer:
Thank you for carefully reading our manuscript and offering us an opportunity to improve our article: Morphological, Ultrastructure, and Chemical Constituents of the Endotesta in Ginkgo biloba L. (plants-2629358). We appreciated the reviewer’s insightful comments and constructive suggestions very much. All these comments/suggestions were highly valued and addressed in this revision. We hope the revised manuscript has now met the standard of the journal. As required, all revisions have been marked using ‘track changes’ so that you can see them in MS Word.
Comment 1: The monthly development scheme should be increased and should contain the time range in numbers, since the scheme should also be applicable to researchers in other regions located in other zones and different climates (this is an international journal).
Response 1: Thank you for advising. The development scheme time range in numbers has been attached. However, we believe there is no necessity to adding development scheme at this time because we collected and dissected our samples according to the developing timing of ginkgo, which were in consistence with previous researches (https://doi.org/10.1111/j.1759-6831.2011.00123.x; DOI: 10.1111/j.1399-3054.2012.01691.x.).
Comment 2: At the end of the introduction, the authors summarize the results of the work, rather than formulating problems and hypotheses to solve or test which the study was aimed at. This paragraph is suitable for a conclusion or discussion and should be reformulated.
Response 2: We have reformulated the results of the work in the last paragraph of introduction.
Comment 3: Figures 5,6,7 should be significantly enlarged and the signatures and the reflection of abbreviations in the signatures and photographs should be checked. They don't match!
Response 3: Thanks for your careful revision. We have enlarged Figures 5, 6, 7 and checked the reflection of abbreviations in the signatures and photographs.
Comment 4: Layers of fabrics must be clearly marked - with color, arrows, etc. Such works are valuable for use and must be done with high quality and not contain discrepancies. The quality of the images should be improved.
Response 4: Thank you for your suggestions. We have marked layers of fabrics with arrows in Figures 5, 6, 7, 8, 9.
Comment 5: Significant morphological deposits, given that the authors paid attention to the composition, should be designated and, if they were not identified, discussed. For example, such deposits are usually characterized as deposits of phenolic compounds. ...Thickening of the cell wall in this layer of cells may be associated with.
Response 5: Thank you for your advices. We have designated the composition and added the caption in Figures 5, 6, 7, all revisions have been marked using ‘track changes’ in manuscript.
Comment 6: Discussion of functional versions of morphological features is required.
Response 6: Thank you for your suggestion. We have discussed the functional versions of morphological features in discussion (3.2) Structural characteristic of the endotesta, such as the dual-layered structure at the micropylar end of the endotesta and the presence of a waxy tumor layer on the surface of the endotesta, provided impermeability to water, thereby restraining transpiration losses (see line 624 - 641).
Comment 7: Figures 8 and 9 do not contain symbols.
Response 7: Thank you for your advice. We have added symbols in Figures 8 and 9.
Comment 8: Figure 10 - very small, no statistics in the caption.
Response 8: We have enlarged Figure 10 and added the statistics in the caption.
Comment 9: The caption of Figure 11 must contain complete information, since the images are viewed separately from the article and must be self-sufficient (object, method, processing method, etc.)
Response 9: Thank you for your suggestion. We have changed the caption of Figure 11 to: Figure 11. Correlation analysis of the four nutrient components. Heat map was created according to the result of Spearman’s correlation analysis of four nutrient components by Original 2023. The data representation of protein includes TAA, EAA, NEAA, EAAI and SRCAA; the UFA and SFA represents fatty acid; vitamins contain VC, VE, VB1, VB2 and VA; and the inorganic elements include K, Ca, Mg, P, Na, Fe, Mn, Cu.

Round 2
Reviewer 2 Report
The manuscript Morphological, ultrastructure, and chemical constituents of the endotesta in Ginkgo biloba L. by Fangdi Li, Ganping Liu, Linying Zhao, Xiaoge Gao, Zhuolong Shen, Fuliang Cao, Qirong Guo examined in detail the issues of ginkgo seed formation. This is quite interesting, since the entire range of uses of this plant is quite wide.
The authors have made a correction to the manuscript; unfortunately, the light microscopy images are very small and it is difficult or impossible to make out the inscriptions on them.
I think the work can be accepted for publication after enlarging the drawings, taking into account the quality of the images and their information content.
Author Response
Response to reviewer
Dear Editor and Reviewer:
Thank you for carefully reading our manuscript and offering us an opportunity to improve our article: Morphological, Ultrastructure, and Chemical Constituents of the Endotesta in Ginkgo biloba L. (plants-2629358). We appreciated the reviewer’s insightful comments and constructive suggestions very much. All these comments/suggestions were highly valued and addressed in this revision. We hope the revised manuscript has now met the standard of the journal. As required, all revisions have been marked using ‘track changes’ so that you can see them in MS Word.
Comment 1: The authors have made a correction to the manuscript; unfortunately, the light microscopy images are very small and it is difficult or impossible to make out the inscriptions on them.
Response 1: Thank you for your advice. We have enlarged the light microscopy images including Figures 1, 2, 3 and 4 so that showing the inscriptions clearly.
